# Synthesis and Electrochemical Performance of Electrostatic Self-Assembled Nano-Silicon@N-Doped Reduced Graphene Oxide/Carbon Nanofibers Composite as Anode Material for Lithium-Ion Batteries

**DOI:** 10.3390/molecules26164831

**Published:** 2021-08-10

**Authors:** Ruye Cong, Hyun-Ho Park, Minsang Jo, Hochun Lee, Chang-Seop Lee

**Affiliations:** 1Department of Chemistry, Keimyung University, Daegu 42601, Korea; cry79838@naver.com (R.C.); rubchem@kmu.ac.kr (H.-H.P.); 2Department of Energy Science and Engineering, Daegu Gyeongbuk Institute of Science and Technology (DGIST), Daegu 42988, Korea; alstkdwh@dgist.ac.kr (M.J.); dukelee@dgist.ac.kr (H.L.)

**Keywords:** lithium-ion battery, silicon nanoparticles, nitrogen-doped graphene, carbon nanofibers, anode material

## Abstract

Silicon-carbon nanocomposite materials are widely adopted in the anode of lithium-ion batteries (LIB). However, the lithium ion (Li^+^) transportation is hampered due to the significant accumulation of silicon nanoparticles (Si) and the change in their volume, which leads to decreased battery performance. In an attempt to optimize the electrode structure, we report on a self-assembly synthesis of silicon nanoparticles@nitrogen-doped reduced graphene oxide/carbon nanofiber (Si@N-doped rGO/CNF) composites as potential high-performance anodes for LIB through electrostatic attraction. A large number of vacancies or defects on the graphite plane are generated by N atoms, thus providing transmission channels for Li^+^ and improving the conductivity of the electrode. CNF can maintain the stability of the electrode structure and prevent Si from falling off the electrode. The three-dimensional composite structure of Si, N-doped rGO, and CNF can effectively buffer the volume changes of Si, form a stable solid electrolyte interface (SEI), and shorten the transmission distance of Li^+^ and the electrons, while also providing high conductivity and mechanical stability to the electrode. The Si@N-doped rGO/CNF electrode outperforms the Si@N-doped rGO and Si/rGO/CNF electrodes in cycle performance and rate capability, with a reversible specific capacity reaching 1276.8 mAh/g after 100 cycles and a Coulomb efficiency of 99%.

## 1. Introduction

Lithium-ion battery (LIB) has high energy density (its volume energy density and mass-energy density can respectively reach 450 W.h/dm^3^ and 150 W.h/kg), high average output voltage (about 3.6 V), and large output power. With its low self-discharge, wide operating temperature range (−30~+45 °C), good environmental compatibility, and long cycle life, it is considered one of the most promising energy storage devices [1,2,3]. However, due to the proliferation of portable electronic equipment, computer equipment, sustainable or hybrid vehicles, and renewable energy storage stations, the demand for higher capacity and longer life batteries has continued to grow with the progress of social development, which leads to the need for higher performance LIBs. Thus, scientists are committed to the development of high-performance electrode materials for LIBs [4,5,6].

Silicon (Si)-based materials are considered to be some of the most promising candidate materials for anode materials for LIBs. Because of their high theoretical specific capacity (~4200 mAhg^−1^), low lithiation potential (~0.4 V vs. Li/Li^+^), rich natural content, low price, non-toxicity, and environmental safety, they are commonly used in lithium-ion anode materials of batteries [7,8]. However, the low conductivity of the active Si material leads to poor electrode rate performance, and the huge volume expansion (about 400%) of the silicon particles during the cycle will lead to problems such as crushing of the electrode material and loss of electronic contact between the particles. This leads to reduced battery efficiency, a shortened life cycle, and breakage of battery cells [9]. The commonly used electrolyte will also form a solid electrolyte interface (SEI) on the silicon surface at a potential of less than 1 V. During the volume change, the SEI can crack and expose the exposed silicon particles, and an increasing amount of SEI is formed on the exposed silicon surface [10]. The SEI film continuously increases the total layer thickness of the silicon particles and quickly fills the electrode holes, thereby preventing the transmission of Li^+^ and electrons, causing the battery capacity to rapidly decrease, and thus limiting the practical application of LIBs in commerce [11].

To solve the above problems, silicon is combined with other materials (e.g., graphene, carbon nanotubes, carbon nanofibers, and other carbon materials) to create a composite material with a stable structure and buffer volume changes to improve conductivity and cycle stability. Graphene has excellent electronic conductivity, good physical and chemical stability, high thermal stability, excellent mechanical flexibility, and high theoretical surface area, and excellent performance, as well as other unique structures. Therefore, it is considered to be an effective coating material for the preparation of lithium-ion batteries. The carbon-carbon bond length of graphene is 0.142 nm. The carrier mobility at room temperature can be as high as 15,000 cm^2^ V^−1^ s^−1^, and its corresponding resistivity is 10^−6^ Ωcm (the lowest resistivity among materials that have been examined in the field) [12,13,14,15]. Graphene-based silicon/carbon composite materials can not only improve the volume change of nano-silicon and form a stable SEI film, but they can also improve the electrical conductivity and lithium storage performance of silicon nanoparticles. However, previous experiments have shown that the agglomeration of graphene particles itself may lead to the poor electrical conductivity of the electrode material, and that it may reduce the stability of the charge/discharge cycle process. Thus, graphene doped with nitrogen atoms has recently attracted substantial interest among scientists. Nitrogen atom-doped reduced graphene oxide (N-doped rGO) is considered to be effective in improving the physical and electrochemical properties of graphene. Nitrogen atoms show more electronegativity than carbon atoms because they have 2 lone pairs of electrons. Thus, the electron density of nitrogen-doped carbon becomes lower and they show stronger electrochemical activity. The electronegativity of nitrogen is stronger than that of carbon, and the hybridization of the lone pair electrons of nitrogen and graphene π system forms a p-π conjugation between the lone pair of electrons of nitrogen and the π electrons of graphene in the plane of graphene, which can improve the charge-transfer capability of N-doped graphene and increases conductivity. The nitrogen atoms will also create a large number of vacancies or defects on the graphite plane, thus providing additional transport channels for the wetting of the electrolyte and the diffusion of Li^+^. Therefore, N-doped graphene is considered to be a promising LiBs material [16,17,18,19,20,21,22]. According to Xing Li et al., the conductivity of the N-doped rGO electrode is enhanced, and the gap between the Si nanoparticles and the N-doped rGO is improved when the self-assembled encapsulation of Si in N-doped reduced graphene oxide is used as an anode material for lithium-ion batteries. Further, the close contact network significantly enhances the electrochemical activity. Concurrently, the uniformly distributed N-doped rGO matrix can effectively buffer the volume change of the Si particles during the repeated lithiation/delithiation process, thereby significantly improving the electrode’s long-term cycle stability [23]. In another study, Ren Na et al. showed that the doping level of nitrogen is controlled by the amount of urea used in the reaction, and that the thickness of the modified layer of Si is controlled by the time of aminopropyltriethoxylsilane (APTES) hydrolysis [24]. However, despite the good electrochemical performance of the Si/N-doped rGO electrode, the increasing diffusion distance of Li^+^ through the graphene interlayer channel with increasing electrode size during charge and discharge still remains a challenge. As a result, there are reductions in the transport capacity of the Li^+^, the conductivity, and the rate performance of the electrode. Further, after multiple charge and discharge cycles, the different volume expansion rates of silicon and graphene may likely lead to the peel-off of Si from graphene, thus resulting in decreased cycle performance.

Carbon nanofiber (CNF) has high heat capacity, chemical stability, high conductivity, good mechanical strength, and a large specific surface area. CNF wraps around the silicon particles, which can not only effectively accommodate and buffer the volume change of silicon, but can also prevent both the electrode structure from cracking and the silicon particles from falling off the carbon base due to the expansion of the surface area [25,26,27]. In addition, the confluence of the silicon nanoparticles, graphene, and carbon nanofibers form a relatively strong three-dimensional structure that can effectively increase the specific surface area of the composite material and stabilize the overall structure of the electrode, thus providing an open channel for the transportation of Li^+^ and electrons [28,29].

In this study, we adopted a synthesis method of self-assembly through electrostatic attraction to prepare a self-supporting Si@N-doped rGO/CNF composite as an anode material, and we were successfully able to fulfill the cost-saving and environment-friendly design concept, achieve a stable cycle of electrochemical materials, and obtain a LIB anode material with excellent mechanical flexibility and lightweight. The modified Si by piranha solution and APTES is more compatible with rGO. N atoms are incorporated into the plane of graphene through a hydrothermal reaction using urea as a precursor. The amount of urea used in the reaction determines the level of nitrogen doping. The sponge-like flexible N-doped rGO, CNF, and Si work together to establish a continuous conductive network in the composite structure that is obtained using a simple technique. This design process ensures the uniform diffusion of the Si nanoparticles in the network formed by rGO, and the introduction of CNF prevents the Si nanoparticles from falling off the graphene sheet, thereby effectively buffering the volume change of the Si nanoparticles. The N-doped rGO also helps improve the conduction of electrons and the diffusion of the Li^+^, thus enhancing the conductivity of the electrode. As a result, it is expected that the cycle stability of the nanocomposite and the rate performance would improve.

## 2. Experimental

### 2.1. Materials and Chemicals

Iron (Ⅲ) nitrate nonahydrate (Fe(NO_3_)_3_·9H_2_O, 98%), copper (Ⅱ) nitrate trihydrate (Cu(NO_3_)_2_·3H_2_O, 99%), aluminum nitrate (Al(NO_3_)_3_·9H_2_O), molybdate ((NH_4_)_6_Mo_7_O_24_·4H_2_O), ammonium carbonate ((NH_4_)_2_CO_3_), hydrogen peroxide (H_2_O_2_, 30%), and urea (CH_4_N_2_O, ≥98%) were purchased from Daejung Chemicals & Metals CO in Korea. All the reagents were of analytical grade and used as received. Silicon nanoparticles (powder, APS ≤ 50 nm, 98%) were purchased from Alfa Aesar, Inc. (Ward Hill, MA, USA). Graphene oxide (GO) was purchased from Angstron materials (Dayton, OH, U.S.A, N002-PS, 0.5%) and used as received. (3-aminopropyl) triethoxysilane (APTES, ≥99%) was provided by AcroSeal in Korea. Ethyl alcohol (anhydrous, 99.9%) and sulfuric acid (H_2_SO_4_, 95–98%) were purchased from Sigma-Aldrich (Burlington, MA, USA). Deionized (DI) water was used in the preparation of all the aqueous solutions throughout the experiments.

### 2.2. Synthesis of Si@APTES and CNF

Figure 1 illustrates the schematic preparation process of Si@N-doped rGO/CNF composite material. First, the Si nanoparticles are diffused in the piranha solution (H_2_SO_4_/H_2_O_2_ = 3:1 *v*/*v*). This process can be used to modify the surface of the Si nanoparticles; that is, the hydroxyl groups are grafted on the surface of the Si nanoparticles to form Si-OH. Next, 30 mL of sulfuric acid solution was added to the beaker, followed by the slow addition of 10 mL of hydrogen peroxide solution to form a homogeneous solution of piranha. Then, 0.2 g of SiNPs was added to the piranha solution, and stirring was continued for 8 h in a water bath at 80 °C. After that, the mixture solution was vacuum filtered and washed with deionized water several times to remove the excess piranha solution on the surface of the SiNPs. Following this washing, the pre-treated SiNPs were dried in a vacuum oven at 60 °C for 24 h. Second, the dried SiNPs were dispersed in 400 mL of DI water, and 8 mL of APTES was added to the solution, and stirring was continued for 24 h. After adding APTES to it and going through the hydrolysis process of APTES, the siloxane group of APTES was easily grafted onto the Si nanoparticles with –OH end caps [30,31,32]. Finally, the obtained SiNPs@APTES solution was washed several times with DI water to remove excess APTES, then placed in a vacuum oven at 60 °C for 24 h.

We used co-precipitation to prepare a bimetallic Fe–Cu (70:30 at.%) catalyst for the synthesis of carbon nanofibers. The Fe–Cu catalyst was used to synthesize carbon nanofibers by chemical vapor deposition (CVD). The synthesis process is illustrated in Appendix A [33].

### 2.3. Synthesis of Si@N-doped rGO/CNF and Si@N-doped rGO

We added 2 g urea (CH_4_N_2_O) to 40 mL GO (0.5%) solution, then mixed for about 12 h until it was completely dissolved. The prepared Si@APTES and GO solution were thoroughly mixed and diffused in the ethanol aqueous solution, and ultrasonic grinding was performed for 4 h. Then, the previously mentioned CNF (0.2 g) was added, and the mixture was ultrasonically milled again for 4 h, as well as continuously stirred for 4 h, so that the substances were fully mixed through physical processes to obtain a highly stable Si@N-doped GO/CNF complex. Thus, the homogeneous solution was transferred into a Teflon-lined stainless-steel autoclave and kept at 180 °C for 24 h. Amino groups (−NH_2_) on the surface of the APTES-functionalized Si nanoparticles can easily bond with the epoxy and carboxyl groups on the GO surface during the hydrothermal process, thus generating electrostatic interactions. Simultaneously, a significant number of N atoms can be introduced into the planar structure of the GO to form different types of N-doped GO configurations during the decomposition of urea. The prepared Si@N-doped GO/CNF dispersion was then vacuum filtered and washed with deionized water several times before the composite was dried in an oven at 60 °C for 24 h. Finally, the obtained Si@N-doped GO/CNF composite was heated to 700 °C at a rate of 10 °C/min in a quartz tube furnace with argon (Ar) gas flow and kept there for 5 h to obtain a thermally-reduced Si@N-doped rGO/CNF composite material. This design process not only ensures that Si nanoparticles can be uniformly diffused in the network formed by rGO, but the introduction of CNF also prevents Si nanoparticles from falling off the graphene sheet, thus effectively buffering the volume change of Si nanoparticles.

To compare the electrochemical performance, another Si@N-doped rGO composite membrane electrode without CNF was prepared as a control sample using the same method. The entire material preparation process is illustrated in Figure 1.

### 2.4. Materials Characterization

The surface morphology and microstructure of the Si@N-doped rGO/CNF and Si@N-doped rGO composite were characterized by field-emission scanning electron microscopy (SEM, S-4800, Hitachi, Tokyo, Japan) and high-resolution transmission electron microscopy (TEM, JEM-2100, JEOL, Tokyo, Japan). The qualitative and quantitative analysis of the elements of the prepared samples was performed by energy-dispersive X-ray spectroscopy (EDX, ARL-3460, Thermo Fisher Scientific, Waltham, MA, USA). The characterizations of the sample composition and crystal structure were performed by taking power X-ray diffraction (XRD) measurements using an Ultima IV, 2 kW system (Rigaku, Toyko, Japan) with Cu-Kα radiation (K = 1.5418 Å), scanned with the 2θ range from 2° to 90°. Raman spectroscopy analysis was performed on a Jobin-Yvon LabRAM HR-800 (Horiba, Kyoto, Japan) with a laser light (λ = 514 nm) in the wavenumber range from 100–3000 cm^−1^. Fourier transform infrared spectroscopy (FTIR) of the sample was performed with a KBr pellet in the frequency range from 4000–500 cm^−1^ using the Nicolet 6700 spectrophotometer (Thermo Fisher Scientific, Waltham, MA, USA). X-ray photoelectron spectroscopy (XPS, Multilab-2000, Thermo Fisher Scientific, Waltham, MA, USA) analysis was performed on a twin anode with Al Kα radiation as an X-ray source. The amounts of SiNPs, N-doped rGO, and CNFs in the composites were measured by thermogravimetric analysis (TGA) using a Diamond TG/DTA Thermal Analyzer (PerkinElmer, Waltham, MA, U.S.A), from 25 °C to 800 °C at a heating rate of 10 °C/min under an atmosphere.

### 2.5. Fabrication of LIBs and Electrochemical Measurements

In this study, two-electrode batteries were prepared using Si@N-doped rGO/CNF and Si@N-doped rGO composite materials as active anode materials for LIBs. To test the electrodes and characterize their electrochemical performances, the working electrodes were prepared by mixing 80 wt% active material, 10 wt% Super P carbon black (conductive agent), and 10 wt% polyvinylidene fluoride (PVDF) binder. Next, the mixture was dissolved in an appropriate amount of N-methylpyrrolidinone (NMP) solvent, which was poured to prepare a mixture of uniformly dispersed negative electrode active slurry. The slurry was then coated on the copper foil current collector and dried at 80 °C for 24 h under a vacuum to form the electrode plate. The electrodes were punched into a negative pole disk with a diameter of 14 mm, and the average load density of each electrode was about 1 mg/cm^2^. Metallic lithium foil as the counter and reference electrodes, and two-electrode lithium-ion coin cells (CR2032) were assembled in a high-purity argon-filled glove box. The separator membrane was Celgard 2600, and the electrolyte was a solution of 1 M LiPF_6_ dissolved in a mixture of ethylene carbonate (EC)/dimethyl carbonate (DMC)/ethyl methyl carbonate (EMC) (1:1:1 by volume). The coin cells’ galvanostatic charge-discharge measurements were tested using a battery tester (Neware Co., Ltd. Shenzhen, China) at the voltage range from 0.01–1.5 V (vs. Li/Li^+^). The specific capacity is calculated based on the whole mass of the anode material. The cyclic voltammetry (CV) was performed with a scan rate of 0.5 mV·s^−1^ between the voltage range from 0.01–1.5 V, at room temperature (25 °C). Electrochemical impedance spectroscopy (EIS) measurements were taken on a CHI 660D electrochemical analysis instrument (CH Instruments, Inc. Shanghai, China) between the frequency range from 100 kHz to 10 mHz at an amplitude of 5 mV.

## 3. Results and Discussion

### 3.1. Structure and Morphology

Figure 2 shows SEM images of Si@N-doped rGO/CNF and Si@N-doped rGO composites and the EDS results of Si@N-doped rGO/CNF. Figure 2a–c shows that the surface of the graphene nanosheets is anchored with a large number of Si nanoparticles, and these Si nanoparticles are not seriously agglomerated but are instead evenly distributed on the graphene sheets. However, we observed different morphologies for Si@N-doped rGO/CNF composites (Figure 2d,e). The Si nanoparticles wrapped in the graphene sheet are tightly entangled by the carbon nanofibers, and the carbon nanofibers are interspersed between the Si nanoparticles to form a sTable 3D structure. The well-spaced structure can effectively adapt to the volume change of the entire electrode structure, thereby reducing electrode pulverization and providing the electrode with excellent mechanical integrity [34]. In addition, the Si nanoparticles are tightly wrapped by wrinkled graphene sheets, which may be due to the introduction of a large number of defects in rGO due to the doping of nitrogen atoms, thus creating more binding sites. This is also the result of a close combination with Si nanoparticles after surface modification. Such an electrode structure will improve its electrochemical performance.

To further elucidate the structure and morphology of the composite material, we performed a TEM image analysis of the composite material (Figure 3). Figure 3a,b shows the TEM image results of the Si@N-doped rGO composite. We can observe that the Si nanoparticles are uniformly diffused on the wrinkled graphene sheet without serious agglomeration, which is consistent with the expected results. Second, as depicted in Figure 3c,d, the Si@N-doped rGO/CNF composites were introduced into CNF; graphene-encapsulated Si nanoparticles are distributed in the cross-linked CNF grid and are tightly entangled by CNF, thus forming a woven structure of graphene and CNF. This effectively prevents Si nanoparticles from falling off the graphene sheet due to volume changes, and it also helps maintain the original electrode structure. This is consistent with the results depicted in the SEM images. In addition, Figure 3e shows that the interplanar spacing of Si nanoparticles is 0.315 nm, and the obvious fringes are identified as Si (111) planes. In Figure 3f, the bright diffraction spots in the three diffraction rings of the selected area electron diffraction (SAED) correspond to the (111), (220), and (311) crystal planes of the Si nanoparticles [35]. At the same time, we provide an analysis of the EDX spectrum results of the Si@N-doped rGO/CNF composite in the supplementary material (Appendix A). As shown in the mapping image, in addition to Si, we also detected C, N, and O in the sample; these mainly come from N-doped rGO and CNF. This shows that N atoms were successfully doped into rGO and tightly combined with the modified Si nanoparticles. This effectively prevents direct contact between the exposed Si nanoparticles and the electrolyte to produce a thicker SEI layer as well as the flexible graphene that can effectively buffer the volume change of the Si nanoparticles in the charging and discharging process, thus improving the electrochemical performance of the composite material.

The X-ray diffraction (XRD) pattern in Figure 4 shows the crystal structure characteristics of GO, rGO, CNF, Si, Si@N-doped rGO, and Si@N-doped rGO/CNF. Si, Si@N-doped rGO, and Si@N-doped rGO/CNF have the same characteristic peaks at 28.4°, 47.3°, and 56.0°, which respectively correspond to the Si (111), (220), and (311) crystal surface (JCPDS 27–1402) [36]. This result is consistent with the results of SAED, and it shows that Si successfully exists in the composite material and that there is no structural change in Si during the process of synthesizing the composite material. In addition, in GO, a strong and narrow representative diffraction peak appears at 2θ = 9.8°, which corresponds to the lattice plane of GO (002). In the curve of rGO, the characteristic peaks of GO at 9.8° and around 20° disappeared, but a new peak appeared near 25°, showing that GO was successfully reduced to rGO in an argon atmosphere at 700 °C [37]. In the Si@N-doped rGO and Si@N-doped rGO/CNF composites, the characteristic peak of rGO also appears near 25°, and its lower strength can be attributed to the relatively higher strength of Si. There are also no other new peaks in the Si@N-doped rGO composite, which shows that no other substances were introduced in the process of self-assembly and nitrogen doping. At the same time, there is also a weak peak near 44° of the Si@N-doped rGO/CNF composite, and this is attribtued to the characteristic peak of CNF.

Figure 4b shows the Raman spectroscopy results of GO, rGO, CNF, Si@APTES/N-doped GO, Si/rGO, Si@N-doped rGO, and Si@N-doped rGO/CNF to further elucidate the microstructure of amorphous carbon. As can be seen in the figure, except for the GO and rGO samples, all other samples exhibit the same characteristic peaks near 287, 515, and 956 cm^−1^, which correspond to the typical Raman mode of crystalline Si [38]. It was further confirmed that the silicon crystal had good crystallinity and did not change during the composite material preparation process. Further, for all samples, two peaks were detected at 1348 cm^−1^ (D band) and 1580 cm^−1^ (G band). Of these, the D band is related to the structural defects and disordered structure in the hexagonal graphite, corresponding to sp3 hybridized disordered carbon, while the G band corresponds to the well-defined sp2 hybridized ordered carbon in the graphite sheet. The relative peak intensity ratio (ID/IG) of the D band and the G band can be used as an important indicator for evaluating the degree of graphitization of carbonaceous materials and the defect density in graphene-based materials. The lower the ID/IG value, the higher the degree of graphitization and the better the conductivity [23,39]. The ID/IG ratios of samples GO, rGO, CNF, Si@APTES/N-doped GO, Si/rGO, Si@N-doped rGO, and Si@N-doped rGO/CNF are 1.05, 0.96, 0.99, 1.03, 1.02, 1.00, and 1.00, respectively (Appendix A). This shows that the Si@N-doped rGO and Si@N-doped rGO/CNF samples have the highest degree of graphitization, and consequently, the best electrical conductivity. Following heat treatment and reduction, the ID/IG value of Si@N-doped rGO is smaller than that of Si@APTES/N-doped GO. This is because most of the oxygen groups removed from the graphene nanosheets and the graphitic carbon structure are reconstructed, which leads to increased disorder of Si@N-doped rGO, and as the degree of carbonization increases, the size of the area in the sp^2^ plane in the plane increases, while the strength of the D band decreases. In comparative samples of Si/rGO and Si@N-doped rGO, due to the introduction of N atoms in the graphitic carbon structure, the defects of graphitic carbon increase; therefore, the ID/IG value of Si@N-doped rGO is smaller than the ID/IG value of Si/rGO [40]. In addition, because CNF itself has some defects, adding CNF will weaken the D band, which will reduce the ID/IG value. These defects can provide more abundant channels for the transportation of Li^+^, reduce the resistance of lithium-ion migration, and improve the electrochemical performance of the composite material [41].

Next, the surface composition and the chemical state of each element are characterized by using XPS to compare Si-OH, Si@APTES, Si@APTES/N-doped GO, Si@APTES/N-doped GO/CNF, Si@N-doped rGO, and Si@N-doped rGO/CNF samples, (Figure 5). Figure 5a shows the obtained whole XPS spectra of Si@APTES, Si@APTES/N-doped GO, Si@APTES/N-doped GO/CNF, Si@N-doped rGO, and Si@N-doped rGO/CNF samples. Si, C, O, and N elements were detected in the sample. Figure 5b shows the high-resolution spectra of Si 2p for each sample. In each case, there are two distinct peaks at 99.5 and 103.2 eV, which are respectively attributed to bulk silicon (Si-Si) and silicon-bonded to oxygen (Si-O). This shows that in the process of material preparation, a small amount of nano-scale silicon powder is exposed to the air, and the surface is oxidized at a certain temperature to produce a small amount of SiO_2_. By comparing the Si 2p spectra of Si@APTES and Si-OH, it can easily be found that the Si-O peak (103.2 eV) has increased, which is attributed to two factors: First, APTES itself contains many Si-O bonds. Second, the surface of the bulk silicon is covered by APTES, which results in the attenuation of the XPS signal of the bulk Si due to the limited probe depth of XPS. In addition, compared to the Si 2p spectra of Si@APTES, it can be seen that the Si-O peaks in the Si 2p spectra of Si@APTES/N-doped GO and Si@APTES/N-doped GO/CNF are increased significantly; this is attributed to the introduction of GO. As shown in Figure 5c, there are three peaks in the C 1s of the Si-OH and Si@APTES samples, which represent C-C at 284.5 eV, C-O/C-N at 286.5 eV, and O-C=O at 288.9 eV. After the addition of GO and CNF, a new peak appeared in the C 1s of Si@APTES/N-doped GO/CNF and Si@N-doped rGO/CNF, which represents C=O at 287.7 eV. Compared to Si@APTES/N-doped GO/CNF, it can be seen that the intensity of the C-C peak in the C 1s of the reduced Si@N-doped rGO/CNF is improved, and the intensities of the C-O/C-N, O-C=O, and C=O peaks are reduced. This is the result of oxygen loss after thermal reduction. Figure 5d shows the N 1s spectra of Si@APTES and Si@N-doped rGO/CNF. Two prominent peaks of 399.0 and 402.5 eV can be observed in the spectrum of Si@APTES, which respectively correspond to the strong hydrogen bonds of amides–NH_2_ and free–NH_2_. Three diffraction peaks at 398.5, 400.6, and 403.9 eV can be observed in the spectrum of Si@N-doped rGO/CNF; these are the characteristic peaks of pyridine N, pyrrole N, and graphite N, respectively. Thus, the content of pyridine N is slightly higher than that of pyrrole N. Pyridine N and pyrrole N may be attributed to the substitution of N atoms for C atoms at the edges or defects of graphene, thereby providing additional channels for lithium-ion diffusion. Graphite N forms a C-N covalent bond by replacing the internal C atoms with N atoms, which is advantageous for improving the conductivity of the graphene sheet.

Figure 6 shows the FTIR spectrum of the relevant sample, which depicts its surface chemical structure. In Figure 6a, the three samples all have strong absorption peaks between 1200 and 1000 cm^−1^, a range which corresponds to the asymmetric stretching and bending of the siloxane group (Si-O-Si) [24]. For pure Si nanoparticles, this is because the surface of Si nanoparticles is oxidized by air to produce a small amount of SiO_2_. This is consistent with the XPS results. Between 1800 and 1600 cm^−1^, there is an absorption peak in each of the three samples, which corresponds to the vibration of H_2_O molecules. It is easy to see that the vibration of H_2_O molecules in Si-OH is stronger, which is due to the presence of hydrophilic–OH groups and the fact that more H_2_O molecules are easily adsorbed on the surface of Si-OH [23,42]. In the Si@APTES sample, there is an absorption peak at 2930 cm^−1^, which is attributed to the vibration of methyl/methylene (–CH2–) in the APTES molecule. After reduction by heat treatment, the peak disappeared. Thus, the broad peaks between 3100 and 3500 cm^−1^ are attributed to some hydrogen bond interactions between the –NH2 groups in APTES and the hydroxyl groups (–OH) on the surface of the Si nanoparticles [43,44]. These observations show that the surface of Si nanoparticles is rich in amino groups. We also found that the Si-OH spectrum has an obvious characteristic absorption peak at 3480 cm^−1^, which is attributable to the vibration of the –OH bond, and that the peak at 3760 cm^−1^ is substantially stronger than the peak in the Si@APTES spectrum. Altogher, we can conclude that APTES has been successfully grafted onto Si-OH.

In the FTIR spectrum of GO (Figure 6b), the absorption peaks at 1715 and 1078 cm^−1^ are respectively attributed to the stretching vibrations of the -C=O bond in -COOH and of the -C-O bond in -C-OH. Comparing the spectra of GO with the other two samples yields that the relative intensity of the absorption peak at 1715 cm^−1^ in Si@APTES/N-doped GO and Si@APTES/N-doped GO/CNF samples is significantly improved, and that the peak at 1645 cm^−1^ disappeared, which is attributed to the stretching vibrations of the –C=O bond and –C=N bond of the amide [45,46,47,48]. Based on the above analysis, we can conclude that a chemical cross-linking reaction with strong hydrogen bond interaction occurred between Si @ APTES and GO.

Figure 6c shows the thermogravimetric analysis curves (TGA) of Si@N-doped rGO/CNF, Si@N-doped rGO, and pure nano-Si in an air atmosphere. In the TGA curve, it can be seen that the weight of the Si@N-doped rGO/CNF composite sample decreased rapidly between 400 °C and 560 °C. This is attributed to the degradation and rapid oxidation of rGO and CNF. From this result, it can be inferred that the weight of the Si@N-doped rGO sample drops rapidly after 550 °C. In the TGA curve of the Si@N-doped rGO/CNF samples, CNF is the first to decompose between 400 °C and 440 °C. However, the weight of pure nano-Si and composite materials significantly increased after 580 °C. This is due to the difficulty of oxidizing the inside of Si nanoparticles as well as the formation of a small amount of SiO_2_ on the surface [34,41,49]. This is consistent with the XPS and FTIR results. Therefore, the weight losses of Si@N-doped rGO/CNF and Si@N-doped rGO samples can be calculated. In the Si@N-doped rGO/CNF sample, the content of N-doped rGO is 13.2%, the content of CNF is 33.4%, and the content of Si nanoparticles is 53.4%. In the Si@N-doped rGO sample, the content of N-doped rGO is 32.3% and the content of Si nanoparticles is 67.7%.

### 3.2. Electrochemical Performance

Figure 7a,b shows the cyclic voltammetry (CV) curves of the coin cell made of Si@N-doped rGO and Si@N-doped rGO/CNF electrodes during the first five cycles, within the potential voltage window of 0.01–1.5 V (vs Li^+^/Li) at a scan rate of 0.1 mVs^−1^. The CV curve of the first cycle of the two electrodes is visually different from the other subsequent cycle curves. In the first cycle of cathodic scanning, a broad and weak cathodic peak appeared near 1.10–1.40 V, which may be attributable to the reaction between the electrode material and the electrolyte as well as the formation of an irreversible SEI film on the electrode surface. However, the cathode peak disappeared in the subsequent cycles, showing that a stable SEI film was formed on the surface of the electrode material after the first cycle [22,23,24]. In addition, the cathode of Si@N-doped rGO/CNF is stronger than that of Si@N-doped rGO at 1.10–1.40 V. This is because the materials introduced with CNF have many defects, which contribute to the electrochemical reaction between the electrode and electrolyte. The strong reduction peak around 0.01–1.2 V corresponds to the amorphous LixSi alloy formed by amorphous silicon during the reversible lithium-ion intercalation/deintercalation process. During the delithiation process, the two oxidation peaks that can be seen at 0.35–0.37 V and 0.52–0.55 V are attributed to the extraction of Li^+^ from the lithium-silicon alloy, while the LixSi alloy decomposes into amorphous silicon [19,49]. Further, as the number of scans increases, the intensity of the anode peak gradually increases as well. This is because the composite electrode material is gradually activated during the cycle, showing that more lithium tends to alloy with Si; these results are consistent with previous reports [34,40,42].

Figure 8a shows a cycle performance comparison of each sample at a current density of 100 mAg^−1^. The charge and discharge capacity value, Coulomb efficiency, and capacity retention rate of each sample are listed in Appendix A. Although both Si@N-doped rGO and Si/rGO/CNF composites have higher initial capacities, with respective values of 3138.8 and 3434.9 mAh/g, the capacity is extremely attenuated after the first cycle, which may be attributed to the fact that the electrode cannot adapt to the change in the volume of silicon particles during the lithiation/delithiation process, and an SEI layer is formed on the surface of the electrode after contact with the electrolyte. However, compared to Si@N-doped rGO and Si/rGO/CNF composite electrodes, Si@N-doped rGO/CNF electrodes show better cycle performance. The initial discharge capacity was 2192.3 mAh/g. After 100 cycles, the capacity retention rate was 58.2% (1276.8 mAh/g). The Coulomb efficiency reached as high as 99% (Figure 8c). The capacity of the Si@N-doped rGO electrode remained at 1091.8 mAh/g after 100 cycles, and the capacity retention rate was 34.8%. The capacity of the Si/rGO/CNF electrode remained at 964.7 mAh/g after 100 cycles, and the capacity retention rate was only 26.5%. The excellent cycle performance of the Si@N-doped rGO/CNF electrode is not only attributed to the surface modification of silicon nanoparticles, which improves the bonding ability between N-doped graphene and silicon nanoparticles, but also to the introduction of CNF, which prevents the silicon nanoparticles from falling off the surface of graphene due to the volume change. The close contact between CNF and graphene and silicon nanoparticles constitutes a 3D cross-linked structure, which can buffer the volume expansion of silicon nanoparticles and effectively suppress the pulsation of silicon nanoparticles caused by huge volume changes. In addition, the presence of many vacancies and defects in N-doped graphene introduces more reaction sites, and N atoms increase the storage capacity of reversible Li^+^, which improves the conductivity of the electrode material. Further, the modified silicon nanoparticles can be uniformly distributed on the N-doped graphene sheet, thereby preventing the aggregation and accumulation of silicon nanoparticles. This ultimately provides more effective channels for the conduction of ions and electrons and promotes the transfer of ions and electrons.

To further illustrate the electrochemical performance of the Si@N-doped rGO/CNF electrode, Figure 8b depicts the rate performance of Si@N-doped rGO and Si@N-doped rGO/CNF electrode under the current density range of 0.1–1 A/g. Compared to the capacity of the Si@N-doped rGO electrode, the Si@N-doped rGO/CNF electrode exhibits a higher capacity at each current density. At the current densities of 0.1, 0.2, 0.5, 1, 0.5, 0.2, and 0.1 A/g, the Si@N-doped rGO electrode after 10 cycles shows respective capacity values of 1368.0, 1233.7, 1170.6, 1137.4, 1180.1, 1218.8, and 1261.8 mAh/g. However, at the current densities of 0.1, 0.2, 0.5, 1, 0.5, and 0.2 A/g, the Si@N-doped rGO/CNF electrode after 10 cycles shows respective capacity values of 1504.0, 1466.2, 1415.7, 1363.0, 1393.9, and 1414.1 mAh/g. When the current density returns to 0.1 A/g again, the reversible capacity can be maintained at 1433.9 mAh/g. The low specific capacity may be a result of the Si nanoparticles being peeled from the graphene sheet due to the shrinkage/expansion of their volume during the charge/discharge process, thus resulting in direct contact between the Si nanoparticles and causing the electrolyte solution to form a thicker SEI film. In addition, the diffusion distance of Li^+^ through the channels between graphene layers will also increase with the superposition of the electrode size, thereby reducing the lithium-ion storage performance of the graphene electrode and affecting the electrochemical performance. By contrast, the excellent rate capability of the Si@N-doped rGO/CNF electrode may be related to the good conductivity of the graphene sheet. The nitrogen atoms used in the graphite plane form a C-N covalent bond in the graphene sheet, thereby changing the electron density of the carbon. The replacement of the edges of carbon atoms by nitrogen atoms also increases vacancies and defects, which further improves the electrochemical lithium-ion storage activity of the composite material. In addition, the addition of CNF can form a relatively strong 3D structure, which can not only effectively accommodate and buffer the volume change of silicon, but also prevent cracking of the electrode structure and prevent silicon particles from falling off the carbon base due to the expansion of the surface area. CNF is interspersed around Si nanoparticles and graphene to reduce the accumulation of graphene layers as well as shorten the transmission distance of Li^+^ and electrons.

To further elucidate the chemical reaction kinetics of each sample, the EIS patterns of different electrodes were studied at frequencies ranging from 10 mHZ to 100 kHZ and at amplitude ratios of 5 mV (Figure 8d). An equivalent circuit for fitting impedance is inserted in Figure 8d, where R_e_, R_SEI,_, and R_CT_ respectively represent the resistance of ion transport in the electrolyte solution, the resistance of Li^+^ migration through the surface membrane, and the resistance of charge transfer. CPE1 and CPE2 correspond to surface film and double-layer capacitors, respectively. We can see that all the curves in the Nyquist plots appear as semicircles in the high-frequency and middle-frequency regions, and as slanted lines in the low-frequency region. The diameter of the semicircle is related to the resistance of lithium ions through the insulating layer on the surface of the active material particles (R_SEI_) and the charge-transfer resistance (R_CT_). The slanted line corresponds to the diffusion resistance of lithium ions within the electrode active material. The diffusion resistance is correspondingly expressed as Warburg impedance (ZW) [24,47,49]. As shown in Figure 8d, the RCT of Si@N-doped rGO/CNF composite material is only 195.0 Ω, which is much lower than the corresponding values of Si@N-doped rGO composite material (295.6 Ω) and Si/rGO/CNF composite material (683.5 Ω). This shows that the Si@N-doped rGO/CNF composite material can effectively promote the conduction of electrons, thereby reducing the charge-transfer resistance and improving the electrochemical performance. To further evaluate whether a stable SEI film was formed, the Nyquist plots of all the samples after 100 cycles were investigated, and the results are shown in Figure 8e. The RSEI (146 Ω) of the Si@N-doped rGO/CNF electrode is slightly smaller than the R_SEI_ (148 Ω) of the Si@N-doped rGO electrode. This indicates that the insertion of CNF is beneficial in minimizing the formation of SEI film on the electrode surface, which consequently reduces the resistance value.

This report also compares SEM images of the Si/rGO/CNF, Si@N-doped rGO, and Si@N-doped rGO/CNF composite electrodes both before cycling and after 100 cycles in the lithiated state; these are shown in Figure 9. After 100 cycles, obvious cracks appear on the surface of the Si/rGO/CNF electrode. This is attributed to the shrinkage and expansion of the Si volume during the repeated lithiation/delithiation process, which causes the activated carbon material to be crushed. The electrode structure is destroyed because Si is easily peeled from the surface of the carbon material during this process. However, compared to the Si/rGO/CNF electrodes, the Si@N-doped rGO and Si@N-doped rGO/CNF electrodes have fewer cracks on the electrode surface after 100 cycles, and the Si@N-doped rGO/CNF electrode has close to a smooth appearance. This is because the three-dimensional structure formed after the introduction of CNF effectively accommodates the volume change of Si and prevents Si from falling off the surface of the graphene due to the volume change, thereby effectively maintaining the integrity of the electrode structure. The surface modification of Si also enhances the bonding ability between N-doped graphene and Si, and Si can be evenly distributed on the N-doped graphene sheet, thus reducing the agglomeration and accumulation of graphene and Si. This provides stronger evidence to further prove that the Si@N-doped rGO/CNF electrode has better electrochemical performance.

## 4. Conclusions

In this study, we successfully synthesized Si@N-doped rGO/CNF composite electrodes through the electrostatic attraction of amino and carboxyl groups and hydrothermal self-assembly. N-doped rGO was successfully introduced into pyridine N, pyrrole N, and graphite N, which made the graphene structure to produce more vacancies and defects. This provided more channels for the transportation of lithium ions and promotes electron transfer during the cycle, thereby improving the conductivity of the electrode. On the one hand, the three-dimensional structure formed by the close combination of Si, N-doped rGO, and CNF can effectively buffer the volume expansion and contraction of silicon nanoparticles, prevent the silicon nanoparticles from peeling off the graphene sheet, and maintain the stability of the electrode; on the other hand, it effectively prevents direct contact between the electrolyte and the silicon nanoparticles, thereby forming a stable SEI film. The Si@N-doped rGO/CNF electrode has a reversible capacity of 1276.8 mAh/g after 100 cycles and a capacity retention rate of 58.2% at a current density of 0.1 A·g^−1^. The composite electrode has excellent cycle stability and rate performance, and it provides a reference value for research into the next generation negative electrodes of lithium-ion batteries.

## Figures and Tables

**Figure 1 molecules-26-04831-f001:**
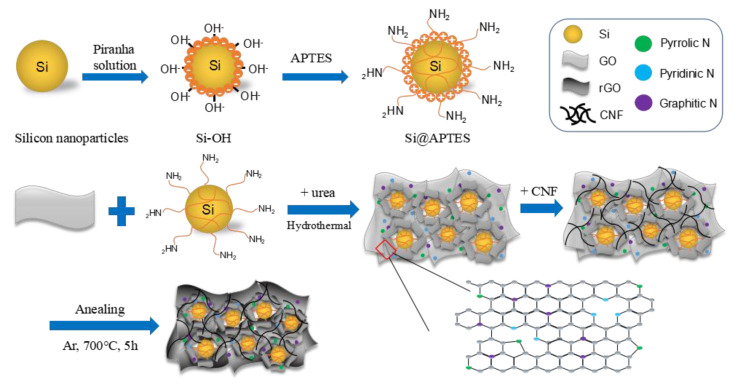
Schematic diagram of the preparation process of Si/N-doped rGO/CNF and Si/N-doped rGO composite material.

**Figure 2 molecules-26-04831-f002:**
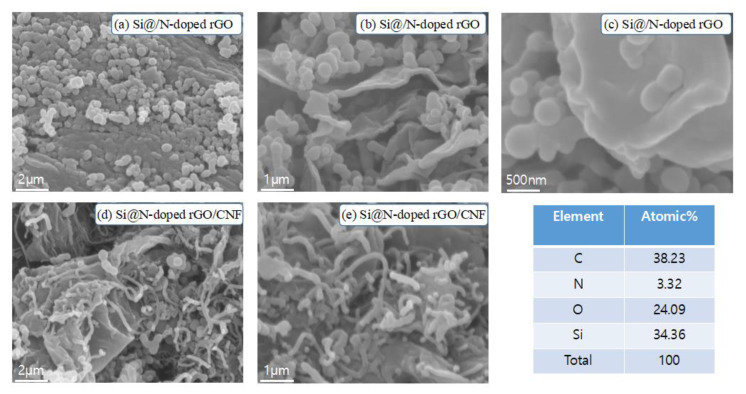
SEM images of the Si@N-doped rGO (**a**–**c**) and Si@N-doped rGO/CNF (**d**,**e**) composites and EDS result of Si@N-doped rGO/CNF.

**Figure 3 molecules-26-04831-f003:**
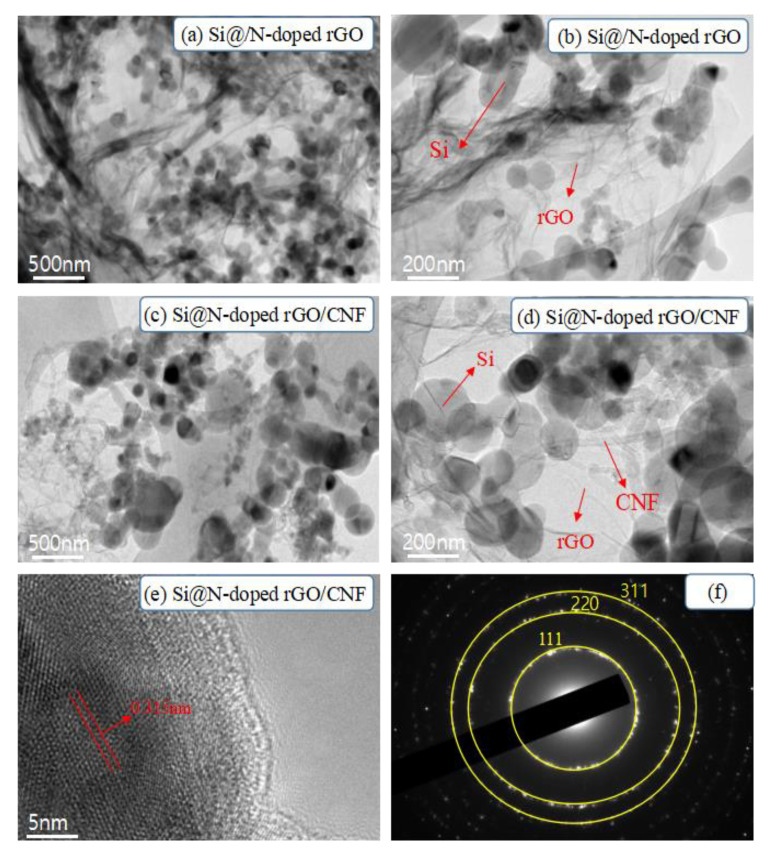
TEM images of Si@N-doped rGO (**a**,**b**), Si@N-doped rGO/CNF (**c**,**d**) composites, (**e**) High-resolution TEM images of Si@N-doped rGO/CNF composites, (**f**) Selected Area Electron Diffraction (SAED) pattern of Si.

**Figure 4 molecules-26-04831-f004:**
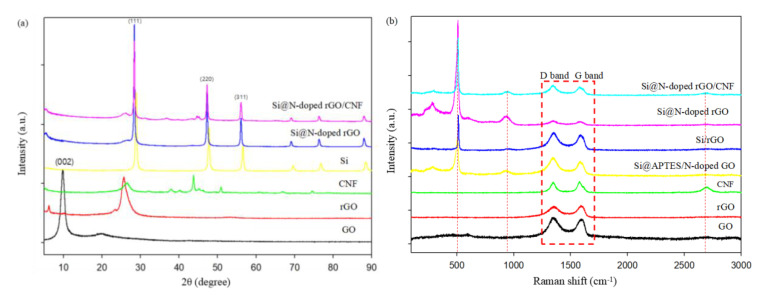
(**a**) X-ray diffraction patterns of GO, rGO, CNF, Si, Si@N-doped rGO, and Si@N-doped rGO/CNF, (**b**) Raman spectra of the GO, rGO, CNF, Si@APTES/N-doped GO, Si/rGO, Si@N-doped rGO, and Si@N-doped rGO/CNF.

**Figure 5 molecules-26-04831-f005:**
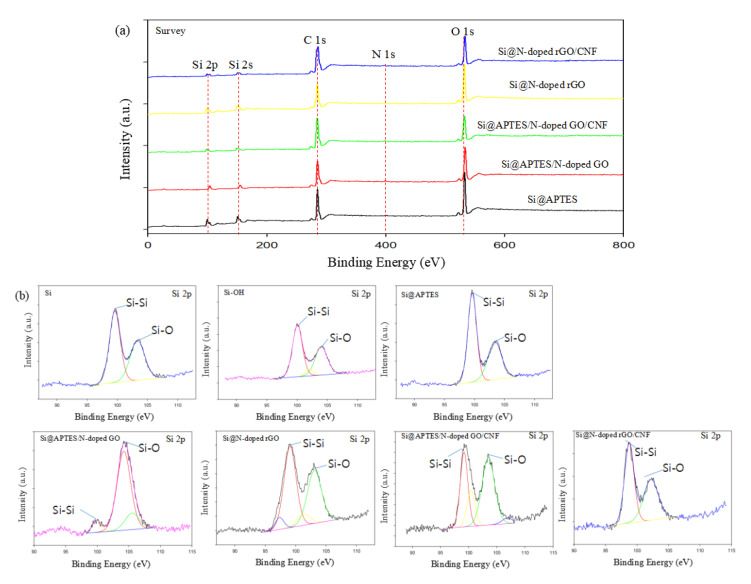
(**a**) Survey XPS spectra of Si@APTES, Si@APTES/N-doped GO, Si@APTES/N-doped GO/CNF, Si@N-doped rGO, and Si@N-doped rGO/CNF, (**b**) Si 2p XPS spectra of Si, Si-OH, Si@APTES, Si@APTES/N-doped GO, Si@APTES/N-doped GO/CNF, Si@N-doped rGO and Si@N-doped rGO/CNF, (**c**) C 1s XPS spectra of Si-OH, Si@APTES, Si@APTES/N-doped GO/CNF and Si@N-doped rGO/CNF, (**d**) N 1s XPS spectra of Si@APTES and Si@N-doped rGO/CNF.

**Figure 6 molecules-26-04831-f006:**
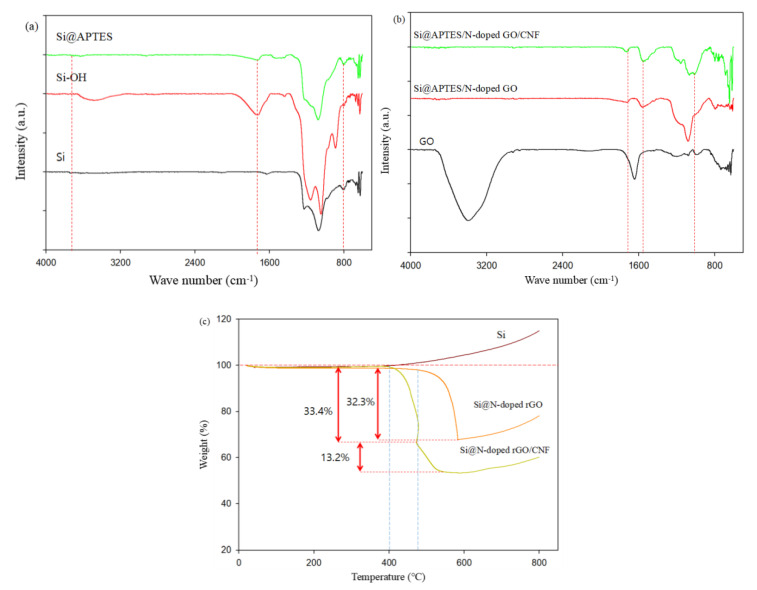
(**a**) Fourier-transform infrared (FTIR) spectra of Si, Si@OH, and Si@APTES, (**b**) Fourier-transform infrared (FTIR) spectra of GO and Si@APTES/N-doped GO and Si@APTES/N-doped GO/CNF, (**c**) TGA curves of the pure nano-Si, Si@N-doped rGO, and Si@N-doped rGO/CNF composite material.

**Figure 7 molecules-26-04831-f007:**
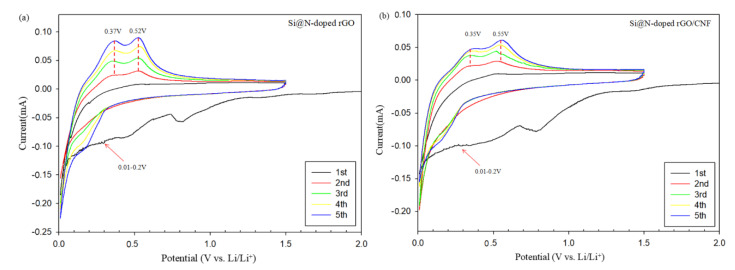
Cyclic voltammetry curves of the synthesized Si@N-doped rGO (**a**) and Si@N-doped rGO/CNF (**b**) electrodes in the initial five cycles.

**Figure 8 molecules-26-04831-f008:**
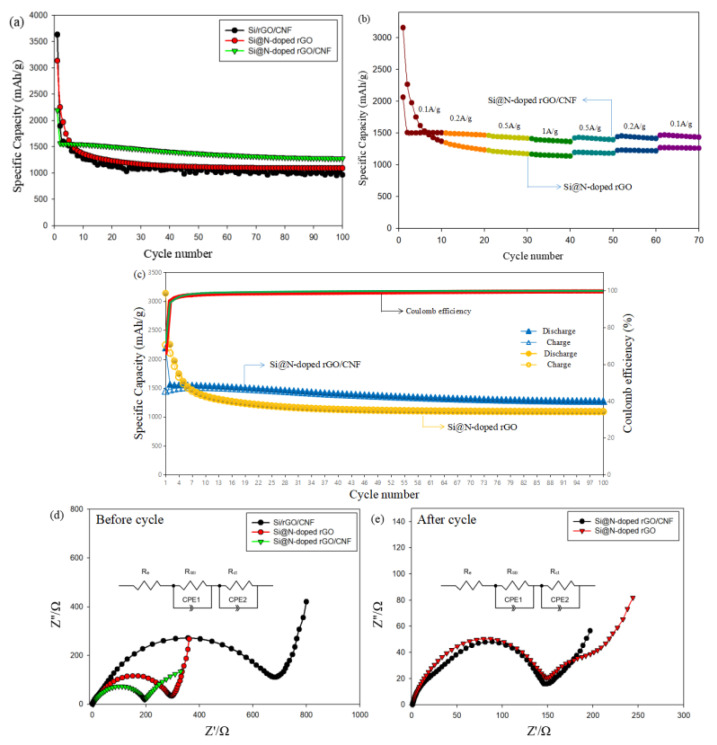
(**a**) Cycling performances of the synthesized Si/CNF/rGO, Si@N-doped rGO, and Si@N-doped rGO/CNF composite electrodes at a current density of 0.1A·g^−1^, (**b**) Rate performances of Si@N-doped rGO and Si@N-doped rGO/CNF electrodes under different current densities, (**c**) Charging and discharging cycle performance and Coulomb efficiency of Si@N-doped rGO and Si@N-doped rGO/CNF electrodes, (**d**) Nyquist plots and electrochemical impedance spectra of the Si/CNF/rGO, Si@N-doped rGO, and Si@N-doped rGO/CNF composite electrodes before the cycle. (**e**) Nyquist plots and electrochemical impedance spectra of the Si/CNF/rGO, Si@N-doped rGO, and Si@N-doped rGO/CNF composite electrodes after the cycle.

**Figure 9 molecules-26-04831-f009:**
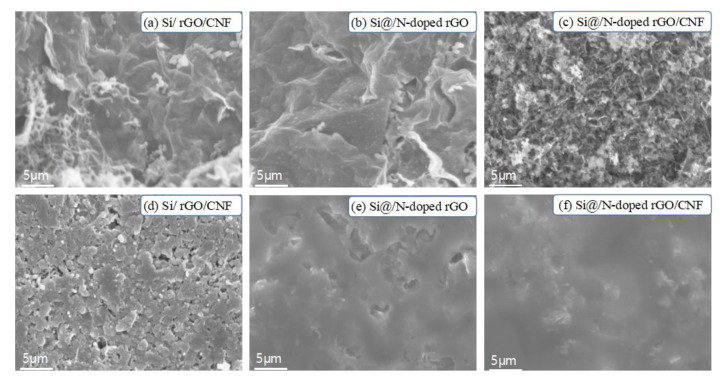
(**a**–**c**) SEM images of the surface of Si/rGO/CNF, Si@/N-doped rGO, and Si@/N-doped rGO/CNF electrodes before the first cycle, (**d**–**f**) SEM images of the surface of Si/rGO/CNF, Si@/N-doped rGO, and Si@/N-doped rGO/CNF electrodes after the 100th cycle at a current density of 0.1 Ag^−1^.

## Data Availability

The datasets generated during and/or analyzed during the current study are available from the corresponding author on reasonable request.

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
