# Peer review of "Synthesis and Electrochemical Performance of Electrostatic Self-Assembled Nano-Silicon@N-Doped Reduced Graphene Oxide/Carbon Nanofibers Composite as Anode Material for Lithium-Ion Batteries"

_molecules, 2021, doi:10.3390/molecules26164831_

Round 1

Reviewer 1 Report

The proposed manuscript entiteled "Synthesis and electrochemical performance of electrostatic self-assembled nano-silicon@N-doped reduced graphene oxide/carbon nanofibers composite as anode material for Lithium-ion batteries" is well written and contains interesting information. The content is clear and the work is well described. The manuscript can be published as presented.

Author Response

To : Reviewer 1,

Thank you so much for your review !

Reviewer 2 Report

This manuscript reported a self-assembly synthesis of Si@N-doped rGO/CNF composites as the anodes for LIBs. The research indicated that this electrode shows good cycle performance and rate capability because the 3D structure can shorten the transfer distance of Li+ and electrons, and this structure exhibits good stability in mechanical properties. Accordingly, this work can be considered for publication in molecules Journal. Other comments are as follows:

  1. The synthesis process is mentioned in introduction part, 2.2, 2.3 and 3.1 repeatedly and tediously. Please make it brief but sufficient.
  2. In figure 4b, please specified the small swell at 300cm-1 for Si@N-doped rGO. Otherwise, please do the experiment again.
  3. From figure 8c, the cycle number of Si@N-doped rGO/CNF is much smaller than Si@N-doped rGO/CNF. Although the capacity retention is better, the conclusion of good cycle performance can not be obtained.
  4. The conclusion in line 465-467 needs more evidence, it cannot be drawn through the cycling performance. The SEM characterization of the anode after cycling is needed.
  5. Please correct the format of reference, for instance ref. 6, 11, 22 and so on.

Author Response

To : Reviewer 2,

Reviewer 3 Report

This study is extensive, uses complex characterization methods and tries to correlate the obtained results. However, the novelty and originality of this work are not clearly explained. The abstract, introduction and results sections must be improved. Please take into account the comments from the attached PDF version of your manuscript. Hopefully it will be easier for you to follow them. Some of the proposed explanations are not sustained by references or can be improved based on the obtained results. There are numerous sentences hard to follow or understand, please rephrase especially where indicated.

Author Response

Response to Reviewer 3 :

Thank you very much for your kind comments on this manuscript. The followings are our responses to your comments.

According to the previous reports, we have known that adding graphene, CNT, or CNF to the silicon-based anode material can improve the conductivity of the active material and maintain the integrity of the electrode structure, thereby improving the cycle performance of the batteries. However, through the previous experimental studies, it has been found that the accumulation of graphene particles itself may lead to the poor electrical conductivity of the electrode material and may reduce the stability of the charge and discharge cycle process. Nitrogen-doped reduced graphene oxide (N-doped rGO) is believed to improve graphene’s physical and electrochemical properties effectively. In addition, although the Si/N-doped rGO electrode has good electrochemical performance, there is still the problem that the diffusion distance of Li+ through the graphene interlayer channel increases with the increase of the electrode size during charge and discharge. This reduces the transport capacity of Li+ and reduces the conductivity and rate performance of the electrode.

Due to the different volume expansion rates of silicon and graphene, Si is very likely to peel off from graphene after multiple charges and discharge cycles, resulting in decreased cycle performance. The carbon nanofibers used in this manuscript have large specific surface area characteristics, high strength, good flexibility, lightweight, good thermal conductivity, high electrical conductivity, and environmental friendliness. The CNF interspersed around the silicon particles effectively accommodates and buffers the volume change of silicon but also prevents the electrode structure from cracking and the silicon particles from falling off the carbon base due to surface area changes. At the same time, N-doped rGO, CNF, and Si NPs work together to establish a continuous 3D conductive network, which can not only effectively increase the specific surface area and provide open channels for electrolyte immersion but also reduce the accumulation of graphene layers and shorten the transmission distance of lithium ions and electrons.

We have rewritten the abstract, introduction and results, and discussion sections and have referenced the comments in the PDF version of the attached manuscript. At the same time, we have also revised one by one according to the content of the comments. For some explanations in the manuscript, we re-searched the relevant references, combined the experimental results obtained to make a more detailed explanation, and added references to the corresponding positions in the manuscript. In addition, we have double-checked the revised manuscript with the help of a professional native speaker.

Round 2

Reviewer 3 Report

The authors improved considerably the manuscript, taking into account the comments and recommendations. However, one phrase still needs to be revised: "Because nitrogen atoms have a lone pairs of electrons and show more electronegativity than carbon atoms."

The word "because" is used when we express the reason or cause of something. In your case, the existence of lone pairs of electrodes is the reason for the better electronegativity of nitrogen. Therefore, a suggestion is to modify in: "Nitrogen atoms show more electronegativity than carbon atoms because they have 2 lone pairs of electrons".